

# Cellular components in tumor microenvironment of neuroblastoma and the prognostic value

Xiaodan Zhong[1,2,3], Yutong Zhang[2], Linyu Wang[1,3], Hao Zhang[1,3], Haiming Liu[1,3] and Yuanning Liu[1,3]

[1] College of Computer Science and Technology, Jilin University, Changchun, Jilin, China
[2] Department of Pediatric Oncology, The First Hospital of Jilin University, Changchun, Jilin, China
[3] Key Laboratory of Symbolic Computation and Knowledge Engineering, Ministry of Education, Jilin University, Changchun, Jilin, China

## ABSTRACT

**Background.** Tumor microenvironment (TME) contributes to tumor development, progression, and treatment response. In this study, we detailed the cell composition of the TME in neuroblastoma (NB) and constructed a cell risk score model to predict the prognosis of NB.

**Methods.** xCell score was calculated through transcriptomic data from the datasets GSE49711 and GSE45480 based on the xCell algorithm. The random forest method was employed to select important features and the coefficient was obtained via multivariate cox regression analysis to construct a prognostic model, and the performance was validated in another two independent datasets, GSE16476 and TARGET-NBL.

**Results.** We found that both immune and non-immune cells varies significantly in different prognostic groups, and were correlated with survival time. The proposed prognostic cell risk score (pCRS) model we constructed can be an independent prognostic indicator for overall survival (OS) and event-free survival (EFS) (training: OS, HR 1.579, EFS, HR 1.563; validation: OS, HR 1.665, 3.848, EFS, HR 2.203, all *p*-values < 0.01) and only independent prognostic factor in *International Neuroblastoma Risk Group* high risk patients (HR 1.339, 3.631; *p*-value 1.76e–2, 3.71e–5), rather than MYCN amplification. Besides, pCRS model showed good performance in grouping, in discriminating MYCN status, the area under the curve (AUC) was 0.889, 0.933, and 0.861 in GSE49711, GSE45480, and GSE16476, respectively. In separating high risk groups, the AUC was 0.904 in GSE49711.

**Conclusion.** This study details the cellular components in the TME of NB through gene expression data, the proposed pCRS model might provide a basis for treatment selection of high risk patients or targeting cellular components of TME in NB.

Corresponding author
Yuanning Liu, liuyn@jlu.edu.cn

# INTRODUCTION

Neuroblastoma (NB) is the third leading malignant disease in children aged 0–14 years, accounts for 7% of pediatric malignant tumors and responsible for 15% cancer-related deaths (*Maris et al., 2007*). The outcomes for NB vary distinctly from case to case; to

be specific, some patients have spontaneous regression without intervention or mild treatment, while others may harbor inferior outcomes even though they have received multimodel therapy, including intensive chemotherapy, surgery, stem cell transplantation, radiotherapy, and molecular therapy (*Pinto et al., 2015*).

The mechanism of spontaneous regression may be induced by the TrkA/NGF pathway, cellular immunity, as well as other alternative mechanisms in NB (*Brodeur & Bagatell, 2014*). Such spontaneous regression can be observed in melanoma, which is tightly related to immune response. *Auslander et al. (2018)* had constructed the predictor called IMPRES based on the theory to predict the response of melanoma to immune checkpoint blocking (ICB) treatment. In their study, the total area under the curve (AUC) was 0.83, which had outperformed the existing predictors. As a matter of fact, the interaction between tumor and its microenvironment affects tumor biological behavior and the associated treatment response (*Joyce & Fearon, 2015*; *Sun, 2016*). Immunocyte infiltration in tumor microenvironment (TME) is tightly correlated with clinical outcomes and treatment response of tumor (*Fridman et al., 2012*). Existing direct and indirect evidence demonstrates that both cells and extracellular matrix in TME contribute to the major hallmarks of tumor in NB (*Borriello et al., 2016*), especially for tumor progression and metastasis.

Studies regarding the effect of cells in TME on NB have been carried out over the past few years. For instance, *Asgharzadeh et al. (2012)* proved that tumor-associated macrophages (TAMs) were more infiltrative in metastatic NB than in local tumors; besides, they suggested that the interactions between tumor cells and inflammatory cells played certain roles in the malignant phenotype and affected tumor prognosis. Additionally, the study by *Hashimoto et al. (2016)* had carried out immunohistochemistry (IHC) on 41 patients and discovered that TAMs and cancer-associated fibroblast (CAF) were markedly correlated with unfavorable clinical outcomes. In addition, *Hishiki et al. (2018)* observed that the invariant NKT cells preferred to infiltrate the spontaneously differentiated and/or regressed NB, and they believed that invariant natural killer T (iNKT) cells potentially contributed to these courses.

Simultaneously, studies on tumor-infiltrating lymphocytes (TIL) have also been conducted. For instance, *Mina et al. (2015)* found that the elevated number of tumor-infiltrating T lymphocytes was correlated with superior survival. *Zhang et al. (2017)* and *Wei et al. (2018)* had analyzed the TARGET-NBL dataset in succession and proposed the immune suppressive microenvironment in NB; nevertheless, the former believed that CD4+ T-cells (Th2 subtype) were related to the favorable survival regardless of the MYCN status, whereas the latter suggested that the enhanced activation of NK cells, cytotoxic T lymphocytes (CTL), and cytolytic signatures were associated with superior survival.

Moreover, some researchers have developed some algorithms to predict the fractions of cell types, especially for immunocytes using transcriptomics data (*Finotello & Trajanoski, 2018*). Among these computational approaches, CIBERSORT (*Newman et al., 2015*) has been widely used to evaluate the immune infiltration of human cancers (*Waks et al., 2019*; *Wei et al., 2018*; *Zhang et al., 2017*; *Zhou et al., 2019*). Typically, CIBERSORT can achieve high accuracy in calculating the immune cell infiltration of human cancers based on the microarray data; however, it can not reflect all cell types of bulk tumors. On the other hand,

the xCell (*Aran, Hu & Butte, 2017*) score algorithm can assess the abundance scores of 64 cell types based on single-cell gene set enrichment analysis (ssGSEA), while the abundance scores can be obtained across both microarray data and RNA-sequencing data.

Therefore, to elucidate the cell components of TME in NB, the xCell score algorithm was employed based on the gene expression data of four independent datasets to portray the cell components within the NB environment. Our results suggested that cell composition was different in NB microenvironment with different prognosis, and cell types were associated with the survival time. Also, a novel model was constructed in this study to provide a more powerful biomarker for predicting the outcomes of NB patients.

## MATERIALS AND METHODS

### Data source and processing

Gene expression profiles and clinical data were downloaded from GEO datasets (GSE49711, GSE45480 (GPL16876), and GSE16476) and the TARGET-NBL data matrix (https://ocg.cancer.gov/programs/target/data-matrix). The survival data of GSE49711 and GSE16476 were acquired from the R2 database (https://hgserver1.amc.nl/cgi-bin/r2/main.cgi). Specifically, a total of 149 NB patients were included in the TARGET-NBL dataset, while 498 were in GSE49711 containing both RNA-seq and clinical data, 649 were included in GSE45480, and 88 were in GSE16476 possessing both microarray gene expression profile and clinical data. Table S1 has summarized the details of baseline patient characteristics in four datasets. GSE49711 and GSE45480 datasets were the training sets, while GSE16476 and TARGET-NBL were the validation cohorts.

### Cell type enrichment abundance analysis based on gene expression data

The cell type enrichment score (xCell scores) was calculated based on the gene expression data using the xCell (*Aran, Hu & Butte, 2017*) web tool (http://xcell.ucsf.edu/). Notably, the xCell tool provides 64 cell types, including immunocytes, stromal cells, stem cells, and other cells; besides, it recruits 10808 genes as the signatures to identify cell types. In this study, the gene names in the four datasets were harmonized with the gene list provided by the xCell tool.

### Subgroup analysis

In view of the clinical data contained included in the four datasets and the known risk factors influencing NB patient survival, patients were classified into high-risk (HR) and non-high-risk (NHR) groups, together with MYCN-amplified (MYCN-A) and MYCN-non-amplified (MYCN-NA) groups. The xCell scores of immunocytes (including activated dendritic cells (aDC), Th cells, conventional dendritic cells (cDC), NKT, Basophils, macrophages M1, and CD8+ T-cells) and non-immunocytes (such as smooth muscle cells, hematopoietic stem cells (HSC), neurons, osteoblasts, endothelial cells, mesenchymal stem cells (MSC), and fibroblasts) in different subgroups were compared and displayed.

### Random forest feature selection and risk score model

Patients were classified into two groups according to progression disease (unfavorable in GSE49711, $n = 91$, and stage 4 in GSE45480, $n = 214$) or regression disease (favorable in GSE49711, $n = 181$, and stage 4s in GSE45480, $n = 78$). Hereafter, the random forest model (*Wang, Yang & Luo, 2016*) was adopted to select cell types for discriminating the two distinct groups, then mean decrease Gini (MDG) and mean decrease accuracy (MDA) was used as the parameters to estimate the importance. Two in three of the samples were randomly selected as training, and the rest were as test to evaluate the importance and the out-of-bag (OOB) value, this step was repeated for 100 times, an average of MDG, MDA, and OOB was as the final result. Simultaneously, the coefficient was obtained through multivariate Cox regression analysis, and the prognostic cell risk score (pCRS) model was constructed (*Zhong et al., 2018*). Patients were classified into high- and low-risk score groups based on the average pCRS value.

### K-means and hierarchical clustering

156 genes of the MYCN-157 signature (*Valentijn et al., 2012*) (FAM85A was missing) were picked up, and k-means clustering was utilized to divide the 498 patients into MYCN_Signagure_Positive (MYCN_Sig_POS) or MYCN_Signature_Negative (MYCN_Sig_NEG) group. In addition, hierarchical clustering was performed to distribute these 498 patients according to the xCell score of 64 cell types, together with the selected parameters 'binary' and 'ward.D2'.

### Statistical analysis

The IBM SPSS version 23, Graphpad Prism 5, and R software were used for all statistical analyses. Kaplan–Meier method and log-rank test were carried out to plot and compare the survival curves. The xCell score or risk score of different groups was compared through the Mann–Whitney U test. A two-tailed *p*-value of <0.05 was considered as statistically significant.

## RESULTS

### Cell components in TME of NB

The final xCell abundance score was tightly correlated with the true cell proportions; as a result, the former was approximated to represent the cell fraction in bulk tumor of NB. It was discovered that in the four NB datasets, immunocytes made up at least one-half of the cell components, followed by stromal cells (Fig. 1A). Compared to MYCN-NA group, MYCN-A patients had fewer immunocytes while more stromal cells. In lymphoid cells, Th and NKT cells had higher xCell scores in most datasets, whereas DC in myeloid cells, especially aDC, had a higher xCell score (Figs. 1B, 1C). Besides, in the four datasets, neurons and stromal cells (such as MSC, smooth muscle cells, osteoblasts, and endothelial cells) also had higher xCell scores (Fig. 1D).

### Subgroup analysis of cell type abundance

Cell types were distinct among different prognostic groups. In GSE49711 and GSE45480, patients were divided into different groups according to the risk factors involved in risk

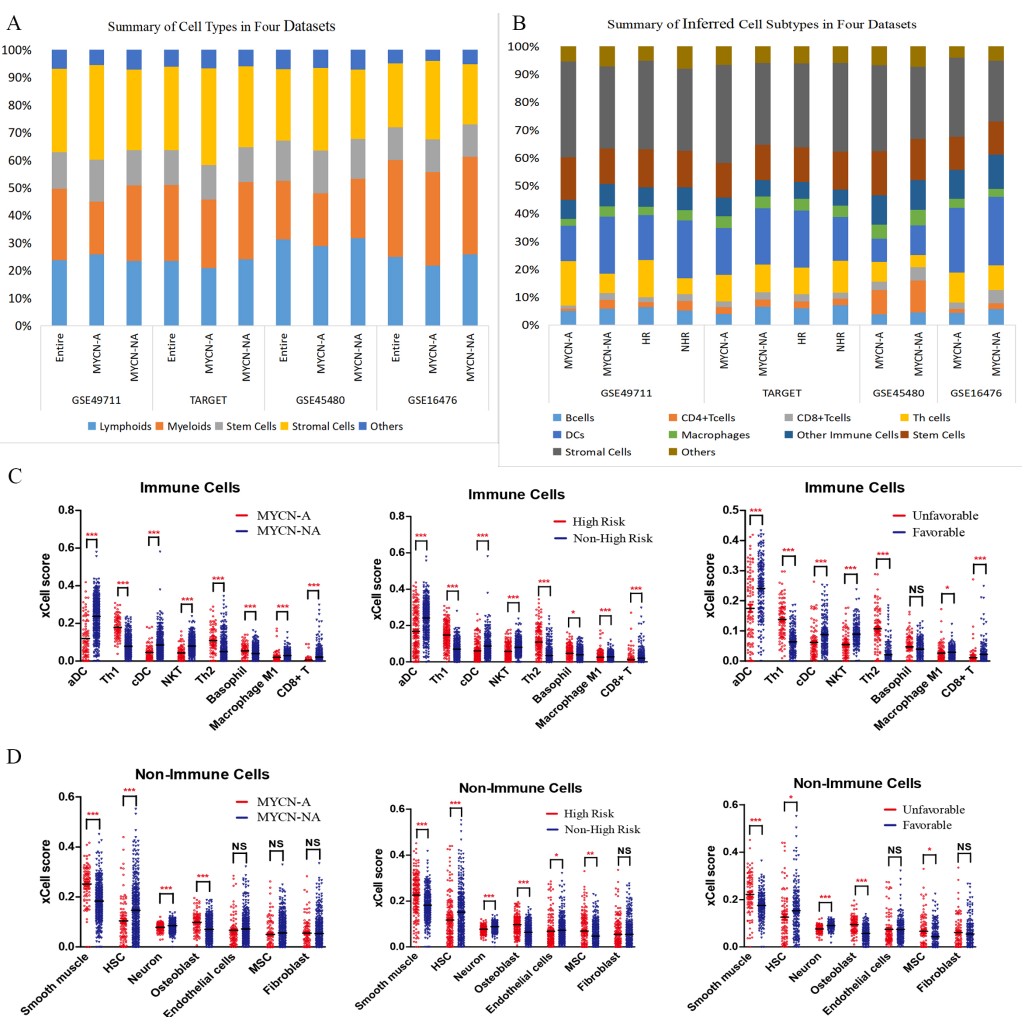

**Figure 1** **Cell types in tumor microenvironment of neuroblastoma.** (A, B) Summary of cell types in four datasets. (C, D) xCell score of immunocytes and non-immunocytes in GES49711. ***denotes $p < 0.001$; ** denotes $p < 0.01$; * denotes $p < 0.05$.

stratification (*Pinto et al., 2015*) of NB, as presented in Figs. 1C, 1D and Fig. S1, Th cells had markedly higher xCell scores in unfavorable prognostic groups; by contrast, DCs, NKT, and CD8+ T cells had much higher xCell sores in favorable prognostic groups. In terms of the stromal cell components, smooth muscle cells and osteoblasts possessed obviously higher xCell scores in worse prognostic groups, while neurons had opposite results. Moreover, HSCs displayed higher xCell scores in favorable prognostic groups. Similar results were found in different INSS stages (Fig. S1C). On the other hand, survival analysis revealed that, the cell abundance of NKT, aDC, cDC, CD8+ T cells, and neurons was positively correlated with long-term survival; while Th1 cells, Th2 cells, smooth muscle cells, and osteoblasts were negatively correlated with survival (Table S2, all *p*-values < 0.0001). We also assessed the correlation of outcomes and Th1/Th2 ratio (after excluding zero value, $n = 368$), and the results of univariate Cox regression analysis showed that Th1/Th2 ratio

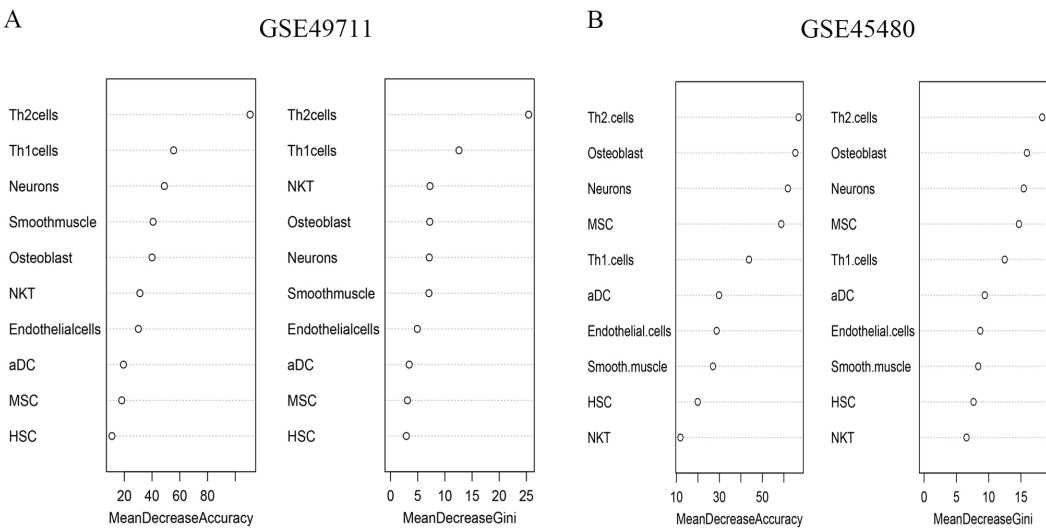

**Figure 2   Importance of ten cell types in random forest algorithm.** (A) MDA and MDG in GSE49711. (B) MDA and MDG in GES45480. MDA, mean decrease accuracy; MDG, mean decrease Gini.

was positively correlated to long-term overall and event-free survival (HR: 0.715 and 0.826, $p$-value: 4.14e−05 and 3.5e−05, respectively).

## pCRS could serve as the independent prognostic factor

The random forest model was employed to select important features. A total of ten cell types were involved after considering the xCell score and the importance of cell types in discriminating groups. After repeating the random forest algorithm for 100 times, the average of the out-of-bag (OOB) value was 11.22% in GSE49711 and 18.32% in GSE45480, respectively. Among them, seven cell types were picked out as the signature involved in the risk score model. The parameter MDG and MDA are shown in Fig. 2. The pCRS model was constructed, and the coefficient was obtained according to the multivariate Cox regression analysis (Table S3). pCRS score = −7.825*NKT+4.942*Th1+2.776*Th2+6.31*MSC-29.11*Neurons+9.285* Osteoblasts+5.185*Smooth muscle. Multivariate Cox regression analysis in GSE49711 confirmed that the pCRS and high risk were the independent prognostic indicators for unfavorable OS and EFS, among which, pCRS had a more significant $p$-value (Table 1 and Table S4). Subsequently, the pCRS model was incorporated into subgroup analysis of GSE49711, including high risk, INSS stage4, age >18 months, and MYCN non-amplified, and the results suggested that patients with higher pCRS value had remarkably shorter survival time in subgroups (all $p$-values <0.001) (Fig. 3).

## Correlation of pCRS with the recognized prognostic molecules and immune markers

Correlation analysis indicated that ERCC6L was positively correlated with pCRS, with the coefficient of up to 0.8112 and the $p$-value <2.2e−16. In addition, the mRNA expression of MYCN oncogene was also positively correlated with pCRS. On the contrary, the expression of UBE4B (*Hugo et al., 2016*; *Zage et al., 2013*), NTRK1 (*Brodeur & Bagatell, 2014*; *Hugo et*

**Table 1  Multivariate Cox regression analysis in training and validation datasets.**

| Parameter | GSE49711 | | | GSE16476 | | | TARGET-NBL | | |
|---|---|---|---|---|---|---|---|---|---|
| | HR | *p*-value | 95% CI | HR | *p*-value | 95% CI | HR | *p*-value | 95% CI |
| pCRS | 1.579 | **4.70E–05** | 1.267–1.967 | 1.665 | **6.79E–03** | 1.151–2.409 | 3.848 | **1.76E–05** | 2.08–7.118 |
| Gender | 0.629 | **0.023** | 0.421–0.938 | 1.147 | 0.71 | 0.558–2.356 | 0.856 | 0.591 | 0.485–1.51 |
| Age group | 1.543 | 0.185 | 0.813–2.930 | 4.638 | **0.027** | 1.193–18.032 | 0.939 | 0.958 | 0.091-9.666 |
| MYCN status | 0.935 | 0.948 | 0.127–6.914 | 1.249 | 0.558 | 0.593–2.632 | 1.639 | 0.157 | 0.826–3.252 |
| High Risk | 4.858 | **7.25E–04** | 1.943–12.149 | NA | NA | NA | 6.191 | 0.156 | 0.497–77.057 |
| INSS stage | 1.202 | 0.203 | 0.906–1.594 | 1.871 | 0.108 | 0.873–4.011 | NA | NA | NA |
| Histology | NA | NA | NA | NA | NA | NA | 1.417 | 0.721 | 0.209–9.594 |
| MKI | NA | NA | NA | NA | NA | NA | 1.236 | 0.254 | 0.859–1.778 |

Notes.

HR, hazard ratio; CI, confidence interval.
Bole type emphasizes $p < 0.05$.

al., 2016), CDH1, and CD44 (*Hugo et al., 2016*), the favorable outcome prognostic factors, was negatively correlated with pCRS (Fig. S2).

Also, cytotoxic T lymphocytes (*Jiang et al., 2018*) (CTL, the average expression of CD8A, CD8B, GZMA, GZMB, and PRF1), and the cytolytic score (*Havel, Chowell & Chan, 2019*) (CYT, the average of GZMA and PRF1) were used to represent the activity of effector T-cells, and their correlation with pCRS was assessed. Our results revealed that pCRS was negatively correlated with the two T-cell indicators ($p < 0.0001$) (Figs. S2G, S2H).

## Comparison with the MYCN-157 signature

MYCN amplification was found in nearly 25% of NB patients, which was associated with unfavorable outcomes (*Huang & Weiss, 2013*; *Maris et al., 2007*). *Valentijn et al. (2012)* found that MYCN non-amplified patients with functional MYCN signature had a poor prognosis, and such phenomenon was confirmed by another study (*Wei et al., 2018*). In this present study, the same results were obtained in GSE49711 (Figs. 4A, 4B). Multivariate analysis suggested that pCRS and MYCN-157 signature could serve as the independent prognostic factors; however, pCRS was more significant (p-value: 7.58e−11 vs. 2.59e−4).

Moreover, hierarchical clustering was also employed to distribute 498 patients into five clusters, and our results indicated that cluster C2 had the worst survival time (Fig. 4C). Meanwhile, the pCRS value of MYCN_Sig_POS and cluster C2 were evidently higher than those of other groups ($p < 0.0001$) (Figs. 4D, 4E).

## Validation in independent datasets

The performance of the pCRS model was validated in another two independent datasets TARGET-NBL and GSE16476. In TARGET-NBL and its high risk group, the age group, MYCN status, histology, MKI, and pCRS were included for Cox analysis, and pCRS was found to be the only independent prognostic factor for OS and EFS (*p*-value:1.76e−5, 9.7e−05 and 3.27e−3. Table 1, Table S5 and Fig. 5). In GSE16476, pCRS and age group were recognized as the independent prognostic indicators for OS (Fig. 6). In the high risk group of GSE49711 and TARGET-NBL, pCRS also serves as the independent prognostic indicator (Table S6). Whereafter, the grouping ability of pCRS was accessed, and the AUC in

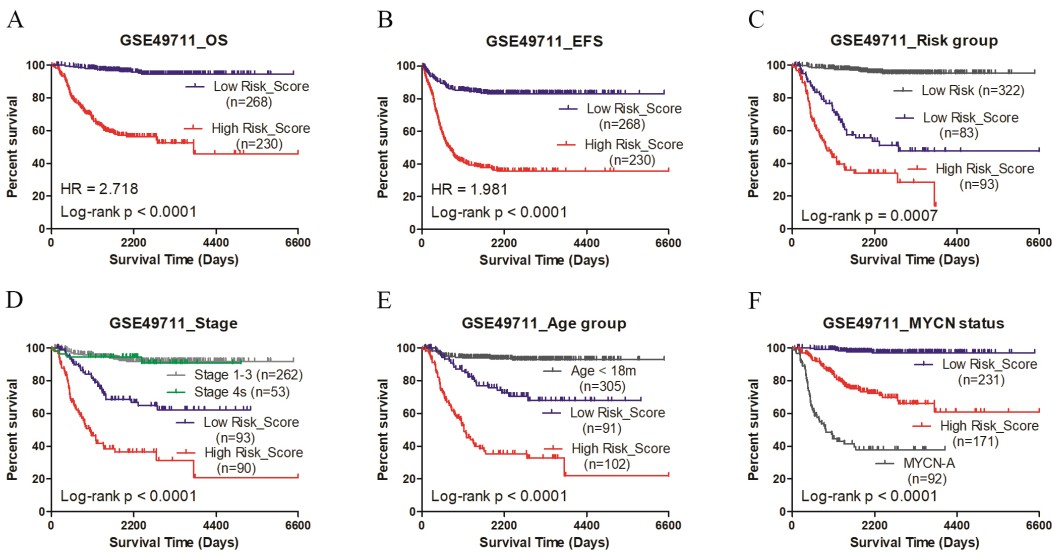

**Figure 3** **Kaplan-Meier plots of high/low pCRS value in GSE49711.** (A) OS. (B) EFS. (C–F) OS in high risk, stage 4, age > 18 months, and MYCN non-amplified. OS, overall survival; EFS, event-free survival.

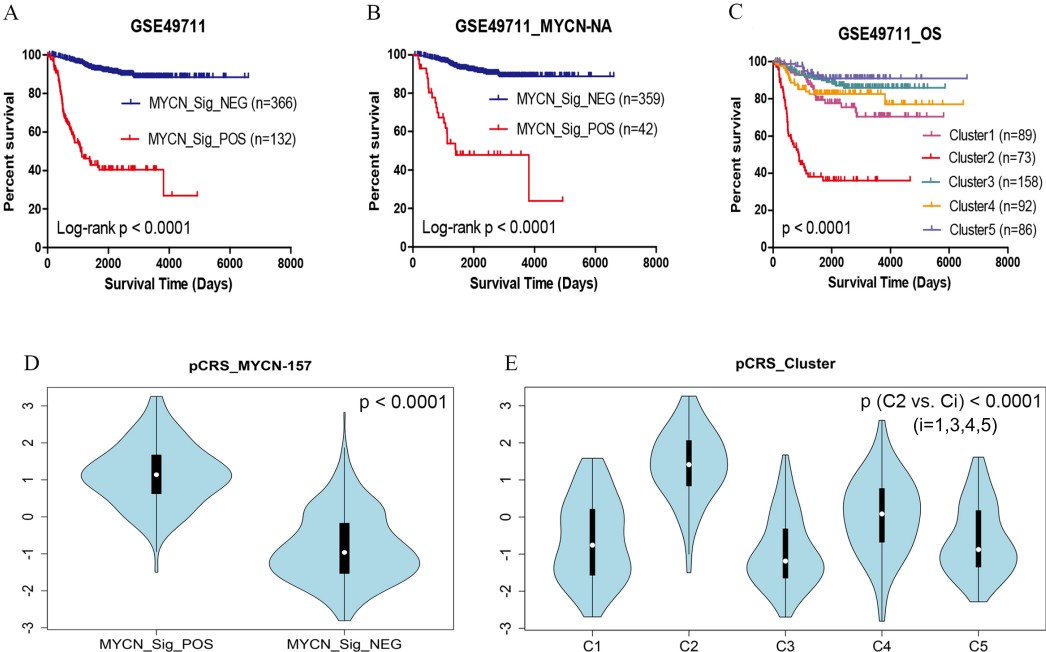

**Figure 4** **OS and pCRS of the MYCN-157 signature and clusters in GSE49711.** (A) OS of MYCN signature positive/negative. (B) OS of MYCN signature positive/negative without MYCN amplified. (C) OS of the clusters. (D, E) pCRS in different groups. OS, overall survival.

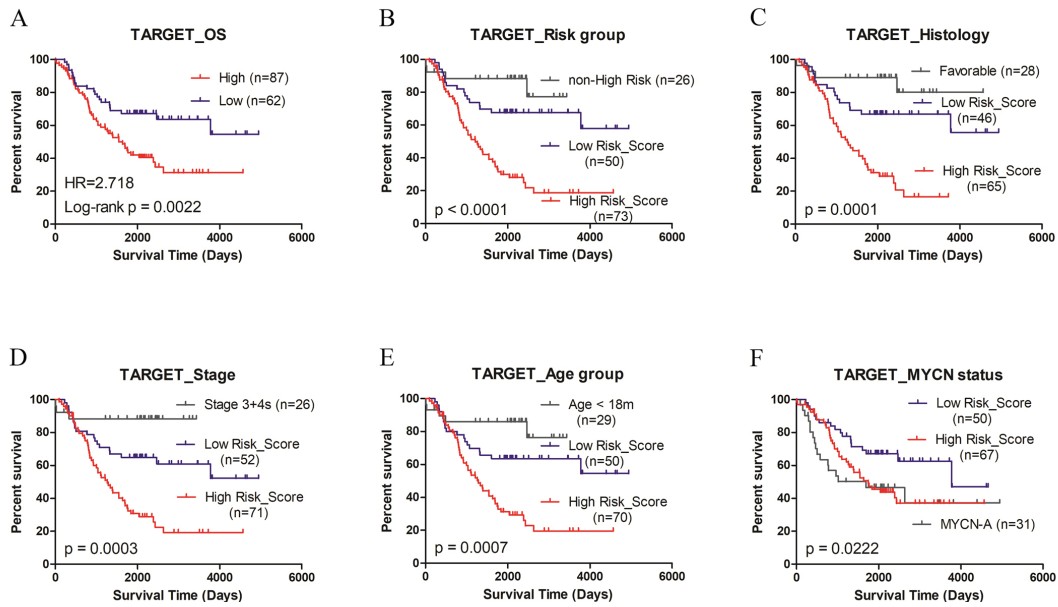

**Figure 5** **Kaplan–Meier plots of high/low pCRS score in TARGET-NBL.** (A) OS. (B–F) OS in high risk, unfavorable histology, stage 4, age > 18 months, and MYCN non-amplified. OS, overall survival.

discriminating MYCN status reached 0.889, 0.933, and 0.861 in GSE49711, GSE45480, and GSE16476, respectively. In addition, the AUC in distinguishing the high risk in GSE49711 was 0.904 (Figs. 7A–7D and Table S7).

Notably, the *p*-value of the xCell was not taken into account in our analysis, which might result in some bias. To confirm the performance of our model, the p-values of the 7 cell types were averaged, and then survival analysis was carried out among the 199 samples, with the mean *p*-value of <0.1. Afterwards, these patients were divided into 4 groups based on the quartile of pCRS, and our results demonstrated that the OS rate and EFS rate gradually decreased with the increase in pCRS ($p < 0.0001$) (Figs. 7E, 7F).

# DISCUSSION

Clinical trials on targeting TME in NB have been carried out and the main strategies focus on TME cells, the signaling pathways, and immunotherapy (*Borriello et al., 2016*). In this study, the cell components in TME of NB were comprehensively estimated through a gene signature-based method named xCell, using wide-ranging in silico analyses from a large set of samples. In addition, both microarray and RNA-seq data were utilized to train and validate the performance of the as-constructed model. Results of xCell score analysis suggested that both immune and non-immune cell components of NB were markedly different among different prognostic groups and that the cell components of TME in NB affected patient survival.

As a matter of fact, studies regarding the influence of immunocytes on NB development and prognostic value have been implemented over the past ten years. However, most of these studies have addressed only one or more of these immunocytes components [9-12],

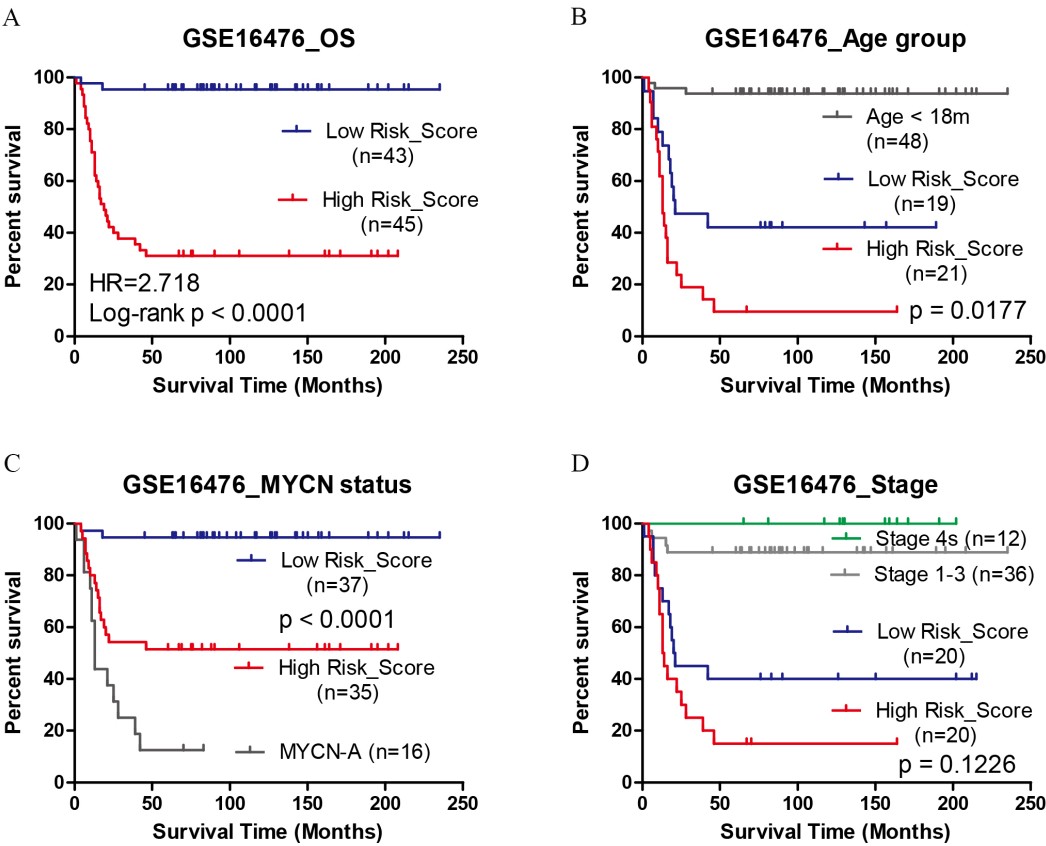

**Figure 6 Kaplan–Meier plots of high/low pCRS score in GSE16476.** (A) OS. (B–D) OS in age > 18 months, MYCN non-amplified, and stage 4. OS, overall survival.

while little attention is paid to the overall cell composition in TME. Furthermore, the prognostic value of Th cells in different adults cancers has also been elaborated (*Fridman et al., 2012*). In this study, we found that both Th1 and Th2 cells were correlated with poor outcomes of NB. As Fridman reviewed, in ovarian cancer, pancreatic cancer, and gastric cancer, Th2 cells infiltration had poor prognosis, and our findings are consistent with them. However, due to the opposite role of Th1 cells in our study, we evaluated the Th1/Th2 ratio in prognostic value, and we found that high Th1/Th2 ratio was correlated with favorable outcomes. We believed that the reason for this result may be that there is some deviation in the mapping of cell components in tumor tissue by gene expression analysis, and the calculation of cell ratios may offset some of this deviation. Therefore, the results need to be further verified by pathological analysis.

There are various cell types in TME of cancers, some cells play bodyguard roles, such as NK cells and CTLs; some cells are "hijacked" and promote cancer progression, like endothelial cells (*Hanahan & Coussens, 2012*), TAMs (*Asgharzadeh et al., 2012*; *Hashimoto et al., 2016*), and CAFs (*Hanahan & Coussens, 2012*). Moreover, some cells, including osteoblasts and MSCs, may enhance bone and bone marrow metastases (*Ma et al., 2011*; *Timaner, Tsai & Shaked, 2019*), which are common in stage 4 NB patients. *Pelizzo et al.*

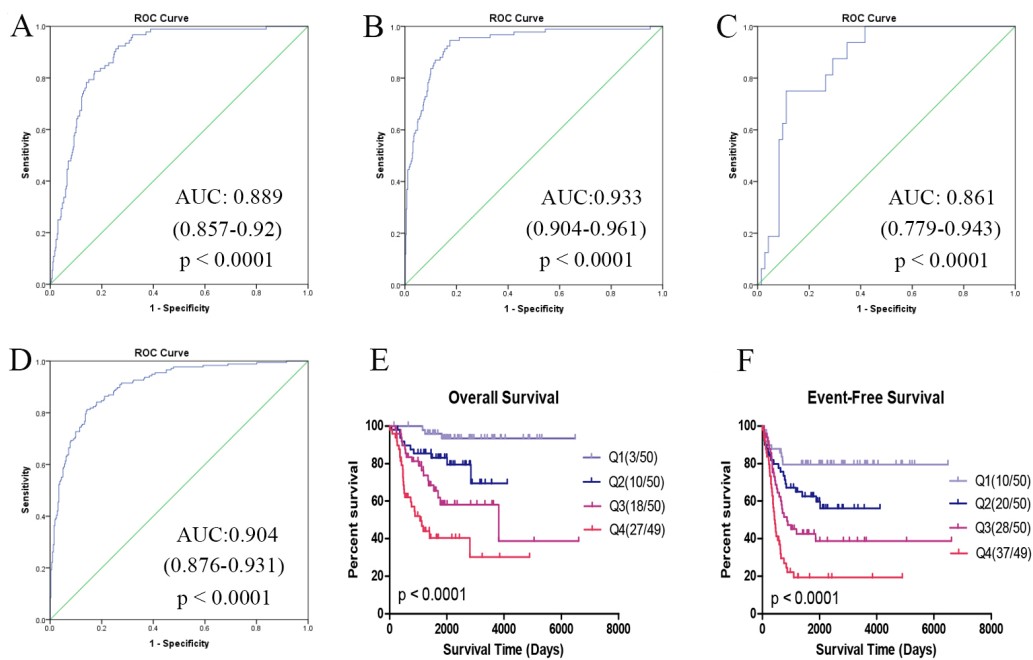

**Figure 7** **Performance in grouping of pCRS model.** (A–C) ROC of discriminating MYCN status in GSE49711, GSE45480, and GSE16476. (D) ROC of grouping high risk in GSE49711. (E, F) OS and EFS of patients considering *p*-value in GSE49711.

*(2018)* had isolated and characterized the tumor-derived MSCs and proposed the potential role of MSCs in promoting tumor invasion and metastasis. Additionally, some cells are tissue-specific, such as epithelial cells clustered in bladder urothelial carcinoma, cervical carcinoma, head and neck squamous cell carcinoma, and other epithelial-derived cancers. Such a phenomenon has been proven by the xCell algorithm (*Aran, Hu & Butte, 2017*). Therefore, neurons are enriched in low-grade glioma and NB, and their roles in tumors are just like what Dirks P.B. said, "neurons refuse cancer's advances" (*Dirks, 2019*). In our study, patients had favorable long-term survival with higher neuron xCell scores in GSE49711.

Cell composition in TME varies depending on the cancer types. In this study, Th cells, DCs, smooth muscles cells, and HSCs accounts for over 40% of all cell types in the TME of NB among the 3 datasets, and we believed that the interactions across different cell types promoted or inhibited tumor progression, thereby affecting tumor prognosis. Hence, a prognostic cell type risk score model (pCRS) was constructed in this study, and the performance of our model was validated in the following analysis.

In the high risk group of INRG staging system (*Pinto et al., 2015*), the current method was unable to distinguish patients with different outcomes, and treatment response and survival time of these patients were inconsistent as well. Our pCRS model was capable of separating these patients with different survival, which might shed some light on treatment regimen selection. The prediction ability of the model was validated in GSE16476 and TARGET-NBL high risk group. Our results showed that the high-risk score group has

the remarkably short OS and EFS, which was superior to that of MYCN amplification. Taken together, the pCRS model might serve as an independent prognostic indicator in all patients and subgroups.

According to our previous studies, ERCC6L might be a cancer-promoting molecule of NB, which was evidently associated with poor prognosis. It was discovered that pCRS was positively correlated with the mRNA levels of ERCC6L and MYCN, whereas negatively correlated with those of favorable prognostic indicator genes, including UBE4B, NTRK1, CHD1, and CD44. CD8+ cytotoxic T lymphocytes play a vital role in eliminating tumors, which lays the foundation of immune checkpoint blocking therapy (*Farhood, Najafi & Mortezaee, 2019*; *Jiang et al., 2018*). Besides, our model was negatively correlated with the cytolytic signatures CYT and CTL. These results are consistent with our expectations.

MYCN-157 signature was identified through silencing MYCN in NB cells, and its high expression was found to be correlated with poor outcomes, regardless of MYCN amplified or non-amplified (*Valentijn et al., 2012*; *Wei et al., 2018*), which was in line with our study. Meanwhile, the MYCN-signature positive group had higher pCRS value than that of the negative group, similar to our speculation. Furthermore, our model was validated in two independent datasets, which could serve as an independent prognostic factor. For INRG high risk patients, our model was the only independent prognostic indicator, which also had good performance in grouping, with the AUC in discriminating MYCN status and high risk of >0.86.

Some limitations should be noted in our study. Firstly, the *p*-value of xCell score was not considered in our analysis because of the remarkably reduced sample size. However, 199 patients with the average *p*-value among seven cell types of <0.1 in GSE49711, and the results showed the survival time was decreased with the increase in pCRS. Noteworthily, the *p*-value of xCell score should be taken into account if more cases can be included in future studies, and more meaningful results may be obtained. Secondly, the results of cell components were derived from in silico analysis, which were not validated via histopathology as a result of the limited experimental conditions and sample acquisition difficulties. This may be taken into account in our future work. Nevertheless, cell composition in TME might be considered for inclusion in risk stratification and decision-making for optimal treatment.

## CONCLUSIONS

Our study portrays the cellular composition in the TME of NB through in silico analysis, providing a clue of targeting cellular components in the TME. The proposed pCRS model might afford some information for the personalized treatment of neuroblastoma patients, especially for high risk patients.

## ACKNOWLEDGEMENTS

The authors would like to thank TCGA and the GEO platform for providing the data.

### Funding

This work was supported by the National Natural Science Foundation of China (No. 61471181), the Industrial Innovation Special Fund of Jilin Province (Nos. 2019C053-2, 2019C053-6). The funders had no role in study design, data collection and analysis, decision to publish, or preparation of the manuscript.

### Grant Disclosures

The following grant information was disclosed by the authors:
National Natural Science Foundation of China: 61471181.
Industrial Innovation Special Fund of Jilin Province (Nos.): 2019C053-2, 2019C053-6.

### Competing Interests

The authors declare there are no competing interests.

### Author Contributions

- Xiaodan Zhong conceived and designed the experiments, analyzed the data, approved the final draft, wrote the first draft of the manuscript.
- Yutong Zhang and Haiming Liu performed the experiments, prepared figures and/or tables, approved the final draft.
- Linyu Wang performed the experiments, authored or reviewed drafts of the paper, approved the final draft.
- Hao Zhang analyzed the data, contributed reagents/materials/analysis tools, authored or reviewed drafts of the paper, approved the final draft.
- Yuanning Liu conceived and designed the experiments, authored or reviewed drafts of the paper, approved the final draft.

### Data Availability

 The raw data of RNAseq and clinical data from TARGET-NBL and clinical data of the GEO dataset are available in the Supplemental Files.

### Supplemental Information

Supplemental information for this article can be found online at http://dx.doi.org/10.7717/peerj.8017#supplemental-information.

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
