# Peer review of "Cellular components in tumor microenvironment of neuroblastoma and the prognostic value"

_PeerJ, doi:10.7717/peerj.8017_

## Round 0.1 · accepted · Accept

Your paper: "Cellular components in tumor microenvironment of neuroblastoma and the prognostic valueis" is acceptable for publication, but please note the reviewers' comments and make the required minor changes while in production as per:

Reviewer 1:

In the present article, authors found that both Th1 and Th2 cells correlated with poor outcomes of NB. This should be further discussed, due to opposite roles of Th1 and Th2 cells in tumor biology and antitumor immunity.

The authors should consider analyzing ratios of immunocompetent cells of interest. For instance, correlation of Th1/Th2 ratio (xCell score) with outcomes of NB should be estimated and discussed.

Reviewer 4:

Please, add some abbreviations missed, as aDc,cDc,iNK, OOB ...etc and place the significance of pCRS abbreviation found in discussion section at the place first time mentioned.

·

Basic reporting

In present study authors present an analysis of cellular makeup of the TME in neuroblastoma (NB) and proposed a cell risk score model to predict the prognosis of NB. Overall the work is very interesting and well-presented. Is it clear and concise paper with precise title, brief and concise abstract and sufficient introduction explaining the idea behind the project. Figures are sharp with legends that explain the figures.The paper is very interesting and the rationale for this work is solid. Presented results are largely novel and the manuscript is suitable for publication in the journal.
Several points:
In the present article, authors found that both Th1 and Th2 cells correlated with poor outcomes of NB. This should be further discussed, due to opposite roles of Th1 and Th2 cells in tumor biology and antitumor immunity.
The authors should consider analyzing ratios of immunocompetent cells of interest. For instance, correlation of Th1/Th2 ratio (xCell score) with outcomes of NB should be estimated and discussed.

Experimental design

no comment

Validity of the findings

no comment

Reviewer 2 ·

Basic reporting

No comment

Experimental design

No comment

Validity of the findings

No comment

Additional comments

The tumor microenvironment (TME) represents a complex network, and plays pivotal role in cancer progression and therapy. In this study, the authors detailed the cellular components of TME in neuroblastoma through gene expression data. Obtained data suggested that pCRS model provide a basis for treatment selection of high risk patients or targeting cellular components of TME in neuroblastoma.
Overall, the manuscript is very interesting and the rationale for this work is solid. The manuscript is recommended for publication.

Reviewer 3 ·

Basic reporting

The authors suggested that cell composition in neuroblastoma microenvironment was markedly different among different prognostic groups and that the cell components of TME in NB affected patient survival.This model was constructed to provide a more powerful predicting of disease outcome of NB patients. Considering morbidity and mortality this investigation is up-to-date. Also, it is well structured, with sufficient field background, easy to read and understand.The literature references are correctly stated.

Experimental design

This is an in silico investigation. Gene expression profiles and clinical data were downloaded from available datasets. Hypothesis were well defined and meaningful, based on literature data. Investigation has been performed on sufficient number of patients, according to high ethical standards, without personal data presentation. Methodology was well described and detailed.

Validity of the findings

It is well known that the microenvironment of the different types of malignant tumors is very powerful predicting factor that can influence disease outcome. In silico analysis are powerful tool, but it is desirable to correlate results with pathohistological and clinical studies to give a more accurate picture of real-life events.

Additional comments

On the basis of all the above mentioned, I consider the paper is suitable for publication in a journal PeerJ.

·

Basic reporting

The reporting is clear, unambiguous, and with professional English language used throughout. The Intro & background is adequate to show the context. The literature is well referenced & relevant. The figures are relevant, high quality, well labelled & described.

Experimental design

Rigorous investigation is performed using a high technical & ethical standard.
Methods are described with sufficient detail & information to be replicate.

Validity of the findings

Meaningful replication encouraged is rationale & benefit to literature is clearly stated.
All underlying data have been provided; they are robust, statistically sound, & controlled.

Additional comments

I commend the authors for their extensive data set, compiled over many years of detailed fieldwork.

Please, add some abbreviations missed, as aDc,cDc,iNK, OOB ...etc and place the significance of pCRS abbreviation found in discussion section at the place first time mentioned.